# Polydopamine Blending Increases Human Cell Proliferation in Gelatin–Xanthan Gum 3D-Printed Hydrogel

**DOI:** 10.3390/gels10020145

**Published:** 2024-02-14

**Authors:** Preetham Yerra, Mario Migliario, Sarah Gino, Maurizio Sabbatini, Monica Bignotto, Marco Invernizzi, Filippo Renò

**Affiliations:** 1Health Sciences Department, Università del Piemonte Orientale, Via Solaroli n.17, 28100 Novara, Italy; 20035225@studenti.uniupo.it (P.Y.); sarah.gino@uniupo.it (S.G.); marco.invernizzi@uniupo.it (M.I.); 2Traslational Medicine Department, Università del Piemonte Orientale, Via Solaroli n.17, 28100 Novara, Italy; mario.migliario@med.uniupo.it; 3Department of Sciences and Innovative Technology, Università del Piemonte Orientale, Viale T. Michel 11, 15121 Alessandria, Italy; muarizio.sabbatini@uniupo.it; 4Department of Health Sciences, Università degli Studi di Milano, Via A. di Rudini n.8, 20142 Milano, Italy; monica.bignotto@unimi.it

**Keywords:** polydopamine, bioprinting, hydrogel, fibroblast, keratinocyte, cell proliferation, skin wound healing

## Abstract

Background: Gelatin–xanthan gum (Gel–Xnt) hydrogel has been previously modified to improve its printability; now, to increase its ability for use as cell-laden 3D scaffolds (bioink), polydopamine (PDA), a biocompatible, antibacterial, adhesive, and antioxidant mussel-inspired biopolymer, has been added (1–3% *v*/*v*) to hydrogel. Methods: Control (CT) and PDA-blended hydrogels were used to print 1 cm^2^ grids. The hydrogels’ printability, moisture, swelling, hydrolysis, and porosity were tested after glutaraldehyde (GTA) crosslinking, while biocompatibility was tested using primary human-derived skin fibroblasts and spontaneously immortalized human keratinocytes (HaCaT). Keratinocyte or fibroblast suspension (100 µL, 2.5 × 10^5^ cells) was combined with an uncrosslinked CT and PDA blended hydrogel to fabricate cylinders (0.5 cm high, 1 cm wide). These cylinders were then cross-linked and incubated for 1, 3, 7, 14, and 21 days. The presence of cells within various hydrogels was assessed using optical microscopy. Results and discussion: PDA blending did not modify the hydrogel printability or physiochemical characteristics, suggesting that PDA did not interfere with GTA crosslinking. On the other hand, PDA presence strongly accelerated and increased both fibroblast and keratinocyte growth inside. This effect seemed to be linked to the adhesive abilities of PDA, which improve cell adhesion and, in turn, proliferation. Conclusions: The simple PDA blending method described could help in obtaining a new bioink for the development of innovative 3D-printed wound dressings.

## 1. Introduction

Polydopamine (PDA) stands out as a remarkable bioinspired biopolymer, drawing inspiration from mussel adhesive proteins and exhibiting a multifaceted array of attributes that render it highly applicable across diverse biomedical domains [1]. Its remarkable biocompatibility, coupled with potent antioxidant, antibacterial, and adhesive properties, positions PDA as a material of choice in diverse applications in the biomedical field [1]. The versatility of PDA is exemplified by its ability to form layers on nearly all types of organic and inorganic substrates, facilitated by the self-polymerization of dopamine (DA) under alkaline conditions [2]. This characteristic enables PDA to serve diverse functions, including tissue adhesion, sealing, surface coating, and biomolecule immobilization, amplifying its utility in biomedical contexts [2]. However, the efficacy of the oxidative polymerization process of DA hinges upon the meticulous control of parameters such as temperature, the pH of Tris-HCl, and the initial dopamine concentration [3]. Dopamine (DA), as the precursor to PDA, emerges as a pivotal material due to its versatility and cost-effectiveness, laying the foundation for the exploration of advanced applications of PDA in biomedical fields [4]. In recent years, PDA-enriched mussel-inspired hydrogels have garnered significant attention for their superior properties compared to conventional hydrogels [5]. These hydrogels, enriched with dynamic catechol-based bonds, exhibit exceptional toughness, extensibility, and a rapid self-healing capacity, marking significant strides in biomaterials engineering [5,6]. Furthermore, PDA modifications in hydrogels allow for increased applications not only in regenerative medicine (increasing cell and tissue adhesion capacity), but also in drug delivery systems [6,7,8,9,10,11]. Notably, PDA-modified hydrogels hold promise as wound dressings, accelerating the healing of chronic wounds [12,13,14]. Bioprinting emerges as a transformative technology in tissue engineering, facilitating the fabrication of intricate, cell-laden 3D scaffolds that mimic native tissue architecture [15,16]. Hydrogel-based bioinks, particularly those incorporating PDA, have emerged as a cornerstone in 3D bioprinting, enabling the precise deposition of cells and biomaterials to generate complex tissue constructs [17,18,19,20]. While the application of 3D bioprinted PDA-modified hydrogels has primarily focused on bone tissue engineering, their versatility suggests potential applications across various tissue types [17,21,22]. In our prior work, we introduced a novel 3D-printable bioink comprising gelatin and xanthan gum, which was successfully employed for printing structures laden with human skin cells [23]. Herein, our investigation revolves around evaluating the impact of incorporating polydopamine (PDA) into the gelatin–xanthan gum hydrogel blend. We aim to ascertain whether this modification enhances the hydrogel’s capacity to facilitate skin cell proliferation while preserving its original physicochemical properties. The ultimate goal is to harness this innovative mussel-inspired hydrogel for dual purposes: serving as a potential wound dressing and as a scaffold for constructing intricate 3D skin-like models.

## 2. Results and Discussion

### 2.1. PDA Does Not Alter Hydrogel Printability and Moisture

The incorporation of Polydopamine (PDA) into Gel–Xnt hydrogel resulted in a dose-dependent darkening of the hydrogel, as depicted in Figure 1A, consistent with expectations [24]. Despite this darkening effect, all PDA-blended hydrogels retained their printability, as evidenced by Figure 1A, indicating that the presence of PDA did not significantly impact the viscosity of the hydrogel. In previous studies, it has been shown that soaking substrates in a dilute aqueous solution of dopamine, buffered to a pH of 8.5, results in the spontaneous deposition of a thin adherent polymer film [25]. This polymer film typically attains a thickness ranging from 10 to 50 nm after 24 h, with its formation appearing independent of the substrate used [25]. In our experimental setup, we introduced a PDA solution into the Gel–Xnt mixture before crosslinking with GTA. Remarkably, this introduction of PDA did not appear to affect the viscosity or crosslinking behavior of the hydrogel, indicating the compatibility of PDA with both Gel and Xnt components.

The main goal of our research on Gel–Xnt printable hydrogels was to use 3D printing technology to fabricate cell-seeded hydrogels for therapeutic purposes (e.g., skin grafts [26], restoring tissue layers [27] or to promote wound healing). Specifically, hydrogels that are employed in wound healing ought to contain a significant water content. This attribute is essential as it enhances cellular interaction and facilitates the diffusion of molecules [28], all while preventing tissue dehydration. 

In CT hydrogels (3% Gel–1.2% Xnt) the percentage of water measured, using the method described by Shawan et al. [28], was 91 ± 0.2% and the blending with different % of PDA did not modify hydrogel water content (Figure 2). In fact, the water percent measured for PDA 1%, 2%, and 3% hydrogels was, respectively, 90.9 ± 0.2%, 91.1 ± 0.3%, and 90.6 ± 1.1% (Figure 2). PDA is an excellent adhesive material with superhydrophilic properties [29], but, in this case, the hydrophilicity of the starting hydrogel was so high that the PDA blending was not able to further increase it.

### 2.2. PDA Blending Effect on Hydrogel Swelling

Evaluating swelling is pivotal in the assessment of hydrogel properties due to its direct correlation with the water absorption capacity, resulting in increased weight and volume [30]. Hydrogels prone to higher degrees of swelling may undergo shape changes and fractures, particularly in moist environments. The swelling behavior of hydrogels can vary based on factors such as gelatin concentration and the presence of glutaraldehyde in their formulation. Glutaraldehyde facilitates the crosslinking of gelatin by reacting with non-protonated ε-amino groups (-NH2) in lysine or hydroxylysine within the gelatin structure, leading to the formation of amide linkages [31]. These crosslinking interactions significantly influence the hydrogel’s water retention capacity [30]. In our CT bioink, gel functional groups effectively bind with all aldehyde groups of the crosslinker, limiting their interaction with water molecules and thereby positively impacting the swelling rate [24]. Figure 3 illustrates that PDA 2% and 3% hydrogels exhibited increased swelling after 1 h (65.5 ± 11.4% and 60.9 ± 7%, respectively) compared to the CT samples (41.4 ± 4.3%, *p* < 0.05), which was attributed to the presence of hydrophilic PDA enhancing the water binding ability. 

However, this increased water binding ability demonstrated by PDA-blended hydrogels diminished after 3 h, as the CT samples showed a swelling ratio of 83.1 ± 6.8%, while PDA 3% exhibited a ratio of 69.0 ± 7%. Similar trends were observed after 6 h (Figure 3). Notably, PDA did not appear to interfere with GTA crosslinking abilities in forming bonds between Gel and Xnt. This suggests a complex interplay between hydrogel components influencing swelling behavior over time. Further investigations are warranted to elucidate the long-term effects and optimize hydrogel formulations for specific applications.

### 2.3. PDA Effect on Hydrogel Hydrolysis

Hydrolysis values were determined following 7 and 14 days of submersion at 37 °C for both CT and PDA-blended printed hydrogels. Despite being predominantly composed of water, hydrogels are susceptible to degradation via hydrolysis, a process where water molecules interact with the hydrogel structure over time, especially under elevated temperatures. The CT hydrogel exhibited a significant percentage of hydrolysis after 7 days (86.1 ± 1.1%), which decreased notably after 14 days (61.4 ± 3.8%) (refer to Figure 4). Interestingly, the incorporation of PDA did not appear to influence the hydrolysis rate of the hydrogel, indicating that PDA does not interfere with the crosslinking process, mediated by GTA, between Gel and Xnt. This observation aligns with our previous findings [23], where we suggested that the sensitivity of CT hydrogel to hydrolysis might stem from the robust crosslinking induced by GTA, resulting in numerous aldehyde groups linked to gelatin via an amide bond, rendering it less stable and more prone to hydrolytic degradation.

### 2.4. PDA Effect on Hydrogel Porosity

Since the ability of cells to proliferate in a 3D structure depends on the pore size and the biodegradability of the structure, it is important to consider the size and number of pores, as well as their geometry and connectivity [32]. Pore sizes should be less than 500 µm to allow for vascularization and tissue formation, as larger pore sizes could decrease cell–cell interactions and, thus, their proliferation [32,33]. In a previously published paper from our lab [23], the porosity of CT hydrogel (3% Gel–1.2% Xnt) was measured using both morphological analysis and a liquid displacement method. The average number of pores measured in an area of 25 µm^2^ was 12,100 ± 561, while their average diameter was 0.723 ± 0.38 µm [23]. The porosity of the CT gel was, therefore, in a range that allows cell survival and communication. Moreover, the percentage of porosity measured using the liquid displacement method for the CT hydrogel was 30.82 ± 4.93% [23], comparable with that measured in the current work (44.1 ± 11.2%,); even the real porosity (average number and size of pores) is reasonable and comparable with that observed in that paper. As shown in Figure 5, PDA blending did not alter the percentage of porosity values which were 43.1 ± 8.2%, 45.3 ± 10.2%, and 46.2 ± 9.4%, respectively, for PDA 1%, 2%, and 3%.

### 2.5. PDA Effect on Cell Proliferation in Cell-Laden Hydrogel 

The important indicators of the cytocompatibility of a biomaterial are cell adhesion and proliferation. We already demonstrated that Gel–Xnt hydrogels were nontoxic and cytocompatible; however, to evaluate the effects of hydrogel functionalization with various concentrations of PDA, human keratinocytes and fibroblasts were mixed with CT and PDA-blended. Cell-laden hydrogels were used to print small cylinders crosslinked using TGA, as described in the Materials and Methods section. Cell-laden cylinders were then washed using PBS and, finally, were cultured for 21 days. Cell growth was monitored b counting the cells present in the hydrogel at different time points. As shown in Figure 6A,B, both keratinocytes and fibroblasts grew slowly until day 7, when, for both cell populations, a significant acceleration in growth in the PDA-blended hydrogel was seen. In fact, on day 7, keratinocyte growth in the CT samples was 120.1 ± 6.9% of T0 (0d), similar to the value scored in PDA 1% samples (123.1 ± 5.7%), while, in PDA 2% and 3%, keratinocyte growth significantly increased, respectively, to 138.2 ± 6.1% and 142.4 ± 3.9% that of CT (*p* < 0.05) (Figure 6A). Keratinocyte growth further increased at day 14, reaching a plateau at day 21 with an ever significant positive difference in PDA-blended hydrogels. In fact, at day 21, keratinocyte growth in the CT samples was 131.8 ± 2.5% of T0 (0 d), while, in PDA 1%, 2%, and 3%, it was, respectively, 141.0 ± 2.1%, 149.2 ± 2.5%, and 156.4 ± 4.7% that of CT (*p* < 0.05) (Figure 6A,B).

A similar cellular growth pattern was observed for human fibroblasts (Figure 7A,B), which, however, grew faster than keratinocytes and reached their maximum growth at day 7. In fact, at day 7, fibroblast growth in the CT samples was 139.5 ± 7.2% of T0 (0 d), while, in PDA 1%, 2%, and 3%, hydrogels fibroblast growth significantly increased, respectively, to 159.7 ± 9.1%, 176.4 ± 9.9%, and 196.1 ± 12.1% that of CT (*p* < 0.05) (Figure 7A). These growth values remained substantially unchanged at day 14 and 21, indicating that fibroblasts reached their maximum space of proliferation (Figure 7A,B)

The faster and higher growth observed for both keratinocytes and fibroblasts in PDA-blended hydrogels compared to the CT samples, even with some important differences due to cell characteristics (epithelial cells vs. connective tissue cells), is believed to be due to the strong adhesion of the cells to the surface of the PDA-mixed hydrogels and the resulting cell survival and proliferation [8]. Of course, a more detailed cell morphological and proteomic analysis is necessary to fully understand the PDA positive effect on cell growth observed in this kind of printable hydrogel. In addition, it may be interesting to evaluate whether a different percentage of PDA or a different quantity of PDA nanoparticles present in a hydrogel [34] could modulate the growth of the different cell population, such as endothelial cells or osteoblasts, to be able, for example, to create a PDA gradient that can act as an “organizer” of small models of complex organs.

Two potential mechanisms of action for PDA in promoting cell proliferation can be postulated. The first, an indirect mechanism, may be linked to the pronounced hydrophilicity imparted by the PDA coating, attributed to its catechol/quinone polar units, amines, and imines [11]. This increased hydrophilicity could play a crucial role in modulating the adsorption of serum proteins on the hydrogel substrate, thereby indirectly influencing cell adhesion and cytoskeletal reorganization, as observed in other scaffold types [35]. Notably, studies have indicated that PDA facilitates the adsorption of proteins while preserving their native configuration, which is pivotal for effective cell adhesion and spreading [36].

However, despite the anticipated increase in hydrogel swelling properties due to the enhanced hydrophilicity induced by PDA, our study revealed no significant rise in hydrogel swelling upon PDA addition. This result could be due to the high hydrogel wettability; consequently, the hydrophilic impact of PDA in our experimental setup may not be paramount. Hence, an alternative direct mechanism of action for PDA in augmenting cell adhesion and proliferation can be proposed. Specifically, this involves the capacity of the PDA coat to elevate the expression of paxillin, a key protein within the focal adhesion complex facilitating cell–substrate anchoring [37,38], as well as inducing the overexpression of the marker MYO, which is responsible for encoding myosin expression in the cytoskeleton. The upregulation of MYO induced by PDA could facilitate the rearrangement of the actin–myosin filaments crucial for establishing robust cell–matrix interactions [38].

## 3. Conclusions

Three-dimensional (3D) bioprinting has emerged as a transformative technology in modern medicine, particularly heralded for its applications in tissue engineering and the treatment of challenging wound conditions. This innovative approach enables the fabrication of complex 3D biological structures that closely mimic the microenvironment of native tissues, fostering crucial interactions between cells and their extracellular matrix. In previous studies, we successfully adapted a Gel–Xnt hydrogel, demonstrating its efficacy as a wound dressing material through rat skin burn experiments [28], and subsequently developed it into a printable bioink [23]. In our latest endeavor, we sought to further enhance the biocompatibility and bioactivity of this hydrogel by incorporating Polydopamine (PDA) during the crosslinking process. PDA, renowned for its natural bioadhesive properties and versatility as a polymer, holds immense promise across various biomedical applications owing to its exceptional chemical adaptability and inherent antioxidant and antibacterial characteristics. Moreover, PDA exhibits the remarkable ability to modulate cell adhesion dynamics and promote cell proliferation at the interface of biomaterials. These attributes position PDA as a compelling biomaterial for facilitating wound healing and skin regeneration. Importantly, our incorporation of PDA into the original Gel–Xnt bioink did not compromise its printability or alter its physicochemical properties, including its moisture content, swelling behavior, hydrolysis rate, and porosity. However, the addition of PDA significantly augmented the growth of human keratinocytes and fibroblasts, suggesting a pronounced enhancement in bioactivity. This positive effect can primarily be attributed to the unique adhesive properties of PDA, which facilitate superior cell adhesion and proliferation within the hydrogel matrix.

Thus, the straightforward method of blending PDA described herein holds promise for the development of a novel bioink with enhanced antibacterial and antioxidant properties, courtesy of PDA’s presence. Such advancements could prove invaluable in the fabrication of next-generation printed wound dressings, offering innovative solutions for improved wound care and tissue regeneration.

## 4. Materials and Methods

### 4.1. Preparation of Polydopamine

Dopamine hydrochloride (2 mg/mL) (Sigma-Aldrich, St. Louis, MO, USA) was polymerized to polydopamine (PDA) in Tris-HCl buffer (pH 8.5) [25]. The prepared dark PDA solution was stored at 4 °C for a maximum period of a month [12].

### 4.2. Preparation of Hydrogel

Shawan et al. [12] laid the groundwork for the preparation of hydrogels targeting wound healing, employing various formulations of gelatin (Gel) and xanthan gum (Xnt). Building upon this research, we developed two novel printable Gel–Xnt hybrid composite hydrogels [23]. For the hydrogel discussed in this paper, we utilized a formulation consisting of 3% Gel and 1.2% Xnt. The synthesis process involved gradually adding bovine Gel (0.6 g) (Sigma-Aldrich, St. Louis, MO, USA) to 20 mL of deionized sterilized water at 70 °C under magnetic stirring to prevent clumping. Subsequently, Xnt (0.12 g) (Sigma-Aldrich, St. Louis, MO, USA) was introduced into the gelatin solution, with a temperature range of 60–70 °C, until water evaporation occurred. The resulting hydrogel (3% Gel-1.2% Xnt) was then stored at 4 °C until use, typically within a week. To incorporate polydopamine (PDA) into the hydrogels, a PDA solution was added drop by drop during mixing to achieve concentrations of 1%, 2%, or 3% (*v*/*v*). Following preparation, the hydrogels were refrigerated for 3 to 10 days and brought to room temperature prior to printing. This meticulous synthesis procedure ensures the stability and consistency of the hydrogel formulations for subsequent applications.

### 4.3. Hydrogel Bioprinting and Crosslinking

CellInk Inkredible 3D printer (CellInk, Gothenburg, Sweden) was used for hydrogels printing. Bio-ink syringes (3 mL) were filled with the different hydrogels and then loaded into the 3D printers (3 mL). Hydrogels were printed with a pressure of 8 kPa to obtain 1 cm^2^ samples. Hydrogels crosslinking was then performed by using 0.3% glutaraldehyde (GTA) (*v*/*v*%), prepared from a 25 *v*/*v*% stock solution. Samples were submerged in GTA solution for 20 min then washed twice with phosphate buffer (PBS) pH 7.4. 

### 4.4. Characterization of 3D-Printed Hydrogel

#### 4.4.1. Moisture

The moisture percentage was calculated according to Shawan’s method [13]. On the third day of stabilization, the hydrated prints were weighed (WH), stored at 37 °C for 2 days to allow the drying process to complete, and then weighed again (WD). The percentage of moisture (water) in a printed hydrogel was calculated by using the following equation: Moisture (%) = [(WH − WD)/WH] × 100
where WH is the original weight of the sample before drying and WD is the weight of the sample after drying. 

All the experiments were replicated three times, with at least three samples for each condition.

#### 4.4.2. Swelling

Swelling test was performed using crosslinked printed hydrogel dried at 37 °C for 48 h. Dried hydrogels were weighed, rehydrated by soaking in deionized water, and weighed again after the residual water was removed by the capillary action of filter paper, at different time points (1, 3, and 6 hs). The swelling ratio (S) was calculated according to Zheng [13].

All the experiments were replicated three times, with at least three samples for each condition.

#### 4.4.3. Hydrolysis

To assess hydrolysis, the cross-linked printed hydrogels were weighed (time 0) and immersed in deionized water at 37 °C. After a period of 4–14 days, the weight was calculated by removing the deionized water and lightly blotting the cross-linked prints with filter paper. The percentage of hydrolysis was obtained with the following equation:Hydrolysis (%) = [(WI − WF)/WI] × 100
where WI is the weight before soaking and WF is the weight remaining after soaking and removal of deionized water. All experiments were replicated three times, with at least three samples for each condition.

#### 4.4.4. Porosity

The porosity of the printed hydrogels was assessed by the liquid displacement method. Absolute ethanol, which causes neither swelling nor shrinking of gelatin [14], was used to immerse the prints. After 5 min of submersion in a known amount of absolute ethanol, the samples were weighed. The porosity of the hydrogel was calculated as follows:Porosity (%) = [(W1 − W3)/ (W2 − W3)] × 100 
where W1 is the initial weight of pure ethanol, W2 is the total weight combining the weight of the hydrogel with that of the ethanol, and W3 is the final weight of ethanol without hydrogel.

### 4.5. Cell Culture

Primary human-derived skin fibroblasts (human fibroblasts were kindly donated by Prof. Marco De Andrea (University of Turin, Italy) and spontaneously immortalized human keratinocytes (HaCaT) (HaCaT were purchased from Cell Lines Service GmbH (Eppelheim, Germany)) were used to test the biocompatibility of the printable hydrogels [23]; Fibroblasts were grown in Petri dishes in RPMI 1640 and keratinocytes in culture flasks in DMEM, both supplemented with 10% heat-inactivated fetal bovine serum (FBS) and 1% Penicillin–Streptomycin and L-Glutamine (Immunological Science, Milan, Italy) in a cell incubator with a humidified atmosphere containing 5% CO_2_ at 37 °C. Small volumes of 100 µL cell suspension containing 2.5 × 10^5^ cells (fibroblast or keratinocytes) were gently mixed, with two syringes connected under a sterile hood. Cylinders, 0.5 cm high and 1 cm wide, were printed in 12-well plates using 12-well model tissue G-code. The cylinders were then crosslinked using GTA. The cell-laden cylinders were washed three times with sterile PBS and incubated for 1, 3, 7, 14, and 21 days in DMEM 10% FBS at 37 °C in a 5% CO_2_ atmosphere. Then, the culture medium was removed and the hydrogels containing the cells were washed 3 times with PBS. Hydrogel images (3 different fields for samples) were digitally acquired using a Zeiss Axiovert 40 cfl (Carl Zeiss Microscopy, LLC, New York, NY, USA) at an original magnification of 20×. The images were analyzed by two different operators blind about the nature of images and the cell number was scored using Image J 1.53m software. Cell number was expressed as % cells present in CT samples at time 0.

### 4.6. Statistical Analysis

Data were presented as mean ± standard deviation (SD). Statistical analyses were performed with GraphPad PRISM software. The Kolgomorov–Smirnoff test was applied in order to understand the normal distribution of data. One-way ANOVA was used for multiple comparisons, while the *t*-Test was performed to compare between the two groups. *p*-values < 0.05 were considered statistically significant.

## Figures and Tables

**Figure 1 gels-10-00145-f001:**
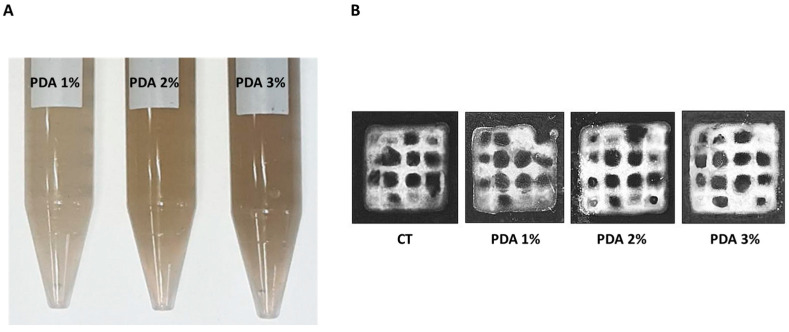
(**A**) Dose-dependent darkening of PDA-blended hydrogels. (**B**) Morphological appearance of CT and PDA-blended hydrogels printed with a pressure of 8 kPa to obtain 1 cm^2^ squares samples crosslinked for 20 min using a solution of 0.3 *v*/*v*% GTA.

**Figure 2 gels-10-00145-f002:**
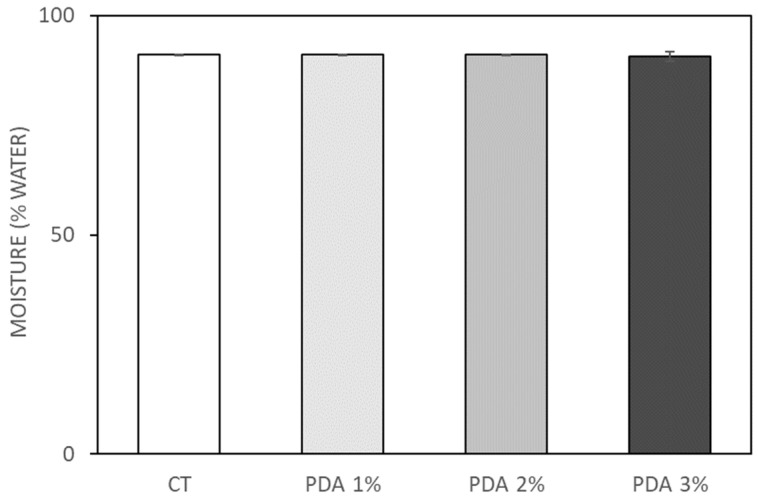
Moisture percentage of CT (Gel3%-Xnt 1.2%) and PDA-blended hydrogels after hydration and drying process at 37 °C for 2 days.

**Figure 3 gels-10-00145-f003:**
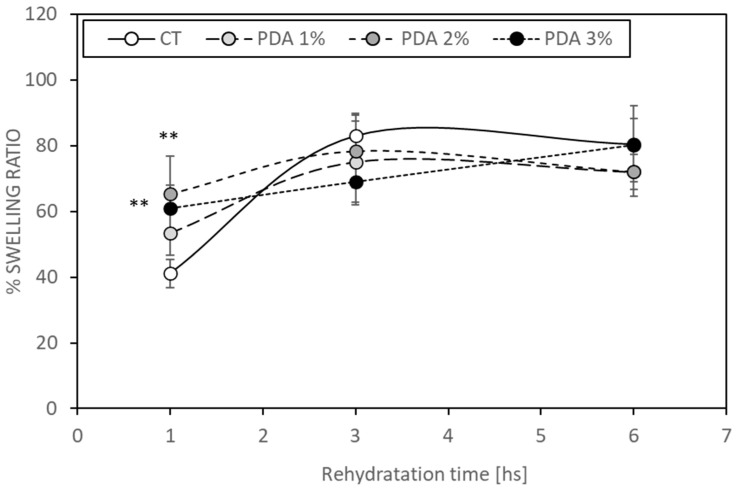
Swelling ratio of rehydrated CT and PDA-blended hydrogels. Dried hydrogels were (1) weighed, (2) rehydrated by immersion in deionized water, and (3) weighed, after the residual water was removed by capillary action of filter paper, at different time points (time 1, 3, 6 hs). ** *p* < 0.05 compared to CT samples.

**Figure 4 gels-10-00145-f004:**
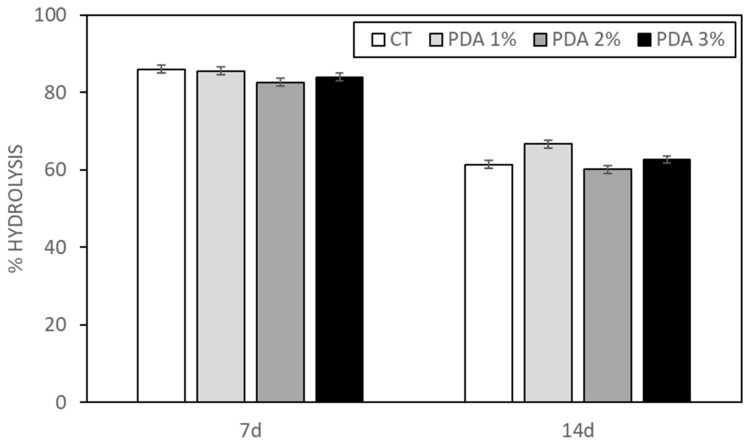
Hydrolysis percentage of CT and PDA-blended hydrogels after soaking in deionized water at 37 °C for 7 and 14 days.

**Figure 5 gels-10-00145-f005:**
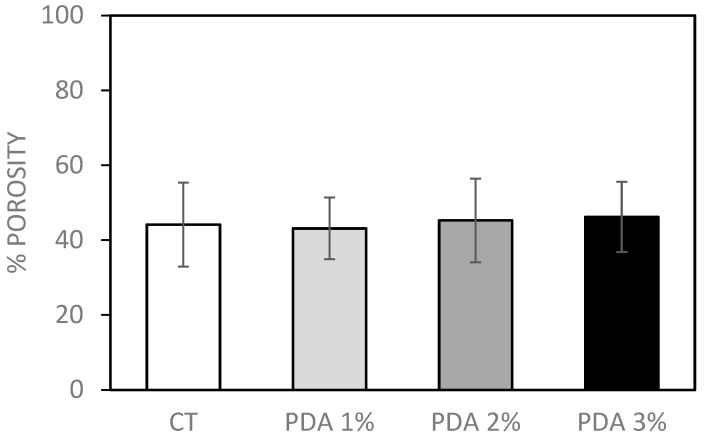
Porosity analysis of CT and PDA-blended hydrogels carried out by liquid displacement method.

**Figure 6 gels-10-00145-f006:**
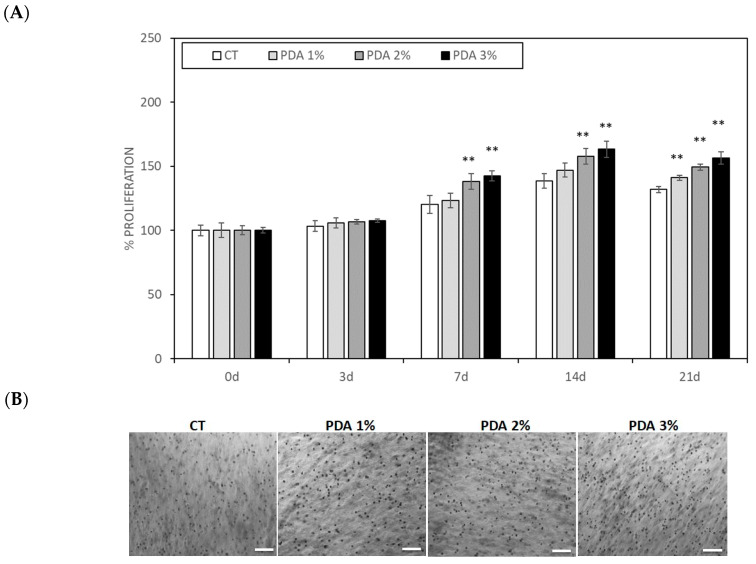
(**A**) Human keratinocytes spontaneously immortalized (HaCaT) proliferation measured in keratinocyte-laden hydrogel printed cylinders. (**B**) Representative images of cell-laden hydrogels after 21 days. Bar = 100 µ. ** *p* < 0.05 as compared to CT samples.

**Figure 7 gels-10-00145-f007:**
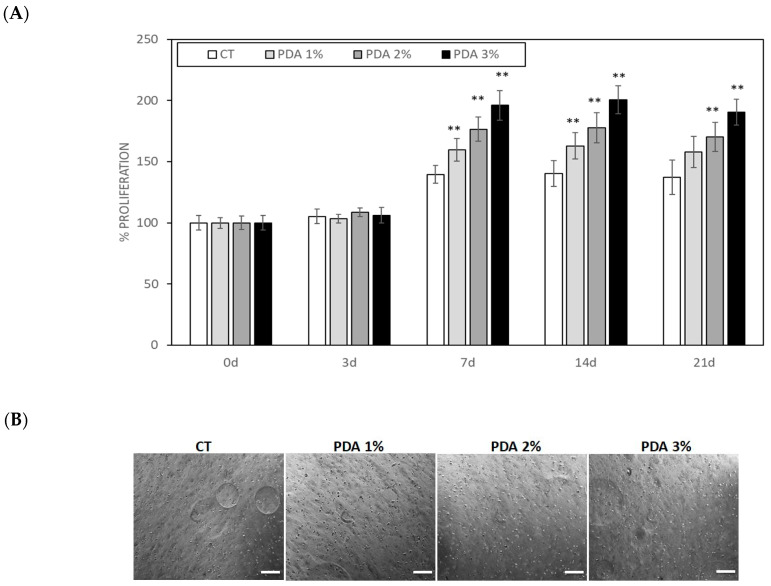
(**A**) Primary human-derived skin fibroblast proliferation measured in cell-laden hydrogel printed cylinders. (**B**) Representative images of fibroblast-laden hydrogels after 21 days. Bar = 100 µ. ** *p* < 0.05 as compared to CT samples.

## Data Availability

The datasets presented in this article are not readily available because the data are part of an ongoing study. Requests to access the datasets should be directed to filippo.reno@unimi.it.

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
