# Peer review of "Polydopamine Blending Increases Human Cell Proliferation in Gelatin–Xanthan Gum 3D-Printed Hydrogel"

_gels, 2024, doi:10.3390/gels10020145_

Round 1

Reviewer 1 Report

Comments and Suggestions for Authors

Overall comments:

The subject matter of this paper deals with the preparation of polydopamine blended gelatin-xanthan gum 3D printed hydrogel to increase human cell proliferation. The study aimed to enhance the properties of Gel-Xnt hydrogel for 3D printing as cell-laden scaffolds by incorporating polydopamine (PDA). The results indicate that PDA blending did not affect hydrogel printability but significantly accelerated fibroblast and keratinocyte growth, suggesting the potential use of this bioink for 3D printed wound dressings. This paper was well-written and can be published after the following minor concerns are revised.

Minor concerns:

Regarding Figure 3 and 6, it is recommended to use a line graph rather than a bar graph to understand the tendency of swelling ratio and % of proliferation to change over time.

The authors mentioned that “In particular, pore sizes should be less than 500 µm to allow for vascularization and tissue formation, as larger pore sizes could decrease cell–cell interaction and, thus, their proliferation." (Page 5, lines 159-161 However, the method used in the experiment was a liquid substitution method, so there is no exact connection with the quote. In the evaluation of pore size, it is recommended to supplement this script by presenting data on hydrogel pore size analysis through quantitative analysis methods such as cryo-SEM, suggesting methods to anticipate specific pore sizes through liquid displacement techniques, or by omitting the sentence itself with citations.

The data presented regarding cell proliferation measurements are indirect and therefore unreliable. To complement this, it is required to present quantification through microscope images or fluorescence images of the cylinder in which cells were cultured.

There is a significant lack of scientific discussion on whether PDA-mixed hydrogel promotes cell proliferation. Rather than simply saying “positive effect,” supplement the script by writing in more detail, such as what chemical mechanism, signaling pathway, or biomolecular mechanism acts to promote cell proliferation, and how PDA has an advantageous effect on creating a microenvironment.

Comments on the Quality of English Language

none

Author Response

The Regarding Figure 3 and 6, it is recommended to use a line graph rather than a bar graph to understand the tendency of swelling ratio and % of proliferation to change over time.

Fig.3 has been changed as request while Fig.6 has not been changed as the line graph appeared not as clear as the bar graph. Moreover fig.6 has been splitted in Fig 6 and 7 adding  light microscopy images of both keratinocyte and fibroblast

The authors mentioned that “In particular, pore sizes should be less than 500 µm to allow for vascularization and tissue formation, as larger pore sizes could decrease cell–cell interaction and, thus, their proliferation." (Page 5, lines 159-161 However, the method used in the experiment was a liquid substitution method, so there is no exact connection with the quote. In the evaluation of pore size, it is recommended to supplement this script by presenting data on hydrogel pore size analysis through quantitative analysis methods such as cryo-SEM, suggesting methods to anticipate specific pore sizes through liquid displacement techniques, or by omitting the sentence itself with citations.

The control hydrogel (3% Gel/1.2% Xant), characterized in a previous paper (Piola eta al. 2022) presented pores...

We add these sentences to the paper in the results and discussion paragraphs (yellow)

The data presented regarding cell proliferation measurements are indirect and therefore unreliable. To complement this, it is required to present quantification through microscope images or fluorescence images of the cylinder in which cells were cultured.

We add some representative  direct light images to show cell presence and density in hydrogels cylinders. We did not use fluorescence images as the hydrogel is slightly autofluorescent

There is a significant lack of scientific discussion on whether PDA-mixed hydrogel promotes cell proliferation. Rather than simply saying “positive effect,” supplement the script by writing in more detail, such as what chemical mechanism, signaling pathway, or biomolecular mechanism acts to promote cell proliferation, and how PDA has an advantageous effect on creating a microenvironment.

We add a part of discussion regarding cell- PDA interactions (yellow)

Reviewer 2 Report

Comments and Suggestions for Authors

Though authors have presented and interesting study on Polydopamine blending increases human cell proliferation in Gelatin-Xanthan gum 3D printed hydrogel their are some concern that needs to be addressed as mention below.

Suggested to remove the full stop from title 

Line no. 17, please correct the spelling of Gelatin-"xantan" gum.

Line no. 24-26, this brief method in abstract is required, suggested to restructure it. "Cell suspension (100 µl, 2.5 × 105 cells) was mixed to uncross linked CT and PDA-blended hydrogel to print cylinders (0.5 cm high, 1 cm wide), cross-linked and incubated for 1, 3, 7, 14 and 21 days. Cells present in different hydrogel was scored by optical microscopy."

Line no. 259 [renò] what does this mean?

Line no. 95 change word "create" to "fabricate"

The discussion section in this manuscript is very week, suggested to improve the section by adding suitable contents following use of proper references.

Thought the results are presented in well framed manner, graphical presentation may loose the reader acceptability. thus suggested to redraw the graphs using stats tool in coloured form, also remove the lower error bars.

The methodology for surface analysis is quite confusing thus make it as simple as possible with clear statement indicating that after hydrogel preparation whether the gels were freeze dried before analysis or something else.

Moreover, briefly elaborate the rheology process.

In addition, to this does author quantify the presence of active constituent, if yes how it was done and if not on what hypothesis the cell culture analysis was performed . how results will indicate the analysis was performed at a concentration were cells were non-toxic or something similar to this.

Author Response

Suggested to remove the full stop from title

Full stop removed

Line no. 17, please correct the spelling of Gelatin-"xantan" gum.

Corrected

Line no. 24-26, this brief method in abstract is required, suggested to restructure it. "Cell suspension (100 µl, 2.5 × 105 cells) was mixed to uncross linked CT and PDA-blended hydrogel to print cylinders (0.5 cm high, 1 cm wide), cross-linked and incubated for 1, 3, 7, 14 and 21 days. Cells present in different hydrogel was scored by optical microscopy."

We restructure the indicated sentence

Line no. 259 [renò] what does this mean?

Sorry, this name was referred to the ref.n. 24 Piola et al. The mistake has been corrected.

Line no. 95 change word "create" to "fabricate"

Changed

The discussion section in this manuscript is very week, suggested to improve the section by adding suitable contents following use of proper references.

We increased the discussion section discussing in a more detailed manner. The added part as been evidenced (yellow)

Thought the results are presented in well framed manner, graphical presentation may loose the reader acceptability. thus suggested to redraw the graphs using stats tool in coloured form, also remove the lower error bars.

Graph n.3 has been changed (suggested by another reviewer) and some representative  images about cell presence in hydrogels. We hope these changes could help in making the article more readable.

The methodology for surface analysis is quite confusing thus make it as simple as possible with clear statement indicating that after hydrogel preparation whether the gels were freeze dried before analysis or something else.

Moreover, briefly elaborate the rheology process.

In addition, to this does author quantify the presence of active constituent, if yes how it was done and if not on what hypothesis the cell culture analysis was performed . how results will indicate the analysis was performed at a concentration were cells were non-toxic or something similar to this.

We do not perform a surface analysis (IR-FTS, XPS or others), therefore no freeze dried samples were prepared.

The aim of our study was to evaluate the effects of blending with PDA on the printability and biocompatibility characteristics of a hydrogel developed starting from a hydrogel designed as wound dressing therefore we did not perform other mechanical tests.

We are sorry, but we are not quite sure to understand the request. However, we do not check for active costituents as PDA behaviour on different surfaces is well known  ( ref. # our paper)  and the fact that cell number did not decreased but increased was an evidence of absence of  PDA toxicity

Round 2

Reviewer 2 Report

Comments and Suggestions for Authors

The present version of the manuscript, reflect the hard effort of the authors to improve the readability and outcome from research envisaged. Thus I recommended for further processing of the manuscript. Thanks